# The Cdc25 and Ras1 Proteins of *Candida albicans* Influence Epithelial Toxicity in a Niche-Specific Way

**DOI:** 10.3390/jof9020201

**Published:** 2023-02-04

**Authors:** Stefanie Wijnants, Jolien Vreys, Jana Nysten, Patrick Van Dijck

**Affiliations:** Laboratory of Molecular Cell Biology, Department of Biology, Institute of Botany and Microbiology, KU Leuven, 3000 Leuven, Belgium

**Keywords:** *Candida albicans*, Cdc25, Ras1, Ras2, PKA, Efg1, Cph1, virulence, epithelial damage, macrophages

## Abstract

The PKA pathway is a signaling pathway involved in virulence in *Candida albicans*. This mechanism can be activated via addition of glucose and activation involves at least two proteins, namely Cdc25 and Ras1. Both proteins are involved in specific virulence traits. However, it is not clear if Cdc25 and Ras1 also affect virulence independently of PKA. *C. albicans* holds a second, atypical, Ras protein, Ras2, but its function in PKA activation is still unclear. We investigated the role of Cdc25, Ras1, and Ras2 for different in vitro and ex vivo virulence characteristics. We show that deletion of *CDC25* and *RAS1* result in less toxicity towards oral epithelial cells, while deletion of *RAS2* has no effect. However, toxicity towards cervical cells increases in both the *ras2* and the *cdc25* mutants while it decreases in a *ras1* mutant compared to the WT. Toxicity assays using mutants of the transcription factors downstream of the PKA pathway (Efg1) or the MAPK pathway (Cph1) show that the *ras1* mutant shows similar phenotypes as the *efg1* mutant, whereas the *ras2* mutant shows similar phenotypes as the *cph1* mutant. These data show niche-specific roles for different upstream components in regulating virulence through both signal transduction pathways.

## 1. Introduction

Morphogenesis in *C. albicans* is controlled by multiple sensing and signaling pathways including the Rim101, the protein kinase A (PKA), and the mitogen-activated protein kinase (MAPK) pathways. These last two pathways are activated by many external stimuli, such as amino acids, CO_2_ levels, quorum sensing molecules, or glucose [1,2,3,4,5]. The MAPK pathway has different parallel pathways which are all activated by specific stimuli, such as cell wall stress and pheromones [6]. One of these mechanisms is the activation of Cdc42 by Ras1, both GTPases. This results in a signaling cascade including Cst20 (MAPKKKK), Ste11 (MAPKKK), Hst7 (MAPKK), and Cek1 (MAPK), and finally the activation of transcription factor Cph1 [6]. Cph1 is involved in hyphae formation on solid medium, mating, and biofilm formation [6,7]. On the other hand, activation of the PKA pathway by glucose is mediated via Cdc25 and Ras1. Activation results in the activation of Cyr1, resulting the production of cAMP [8,9]. cAMP then binds to the regulatory subunits of PKA, releasing the catalytic subunits which then phosphorylate different effector proteins, including the transcription factor Efg1 [1,10]. Expression of Efg1 results in morphogenesis and the expression of various hydrolases [7,11]. Since PKA is not only involved in morphogenesis, but also influences other virulence traits, such as e.g., white-opaque switching or expression of several cell wall proteins, it is an important signaling pathway for *C. albicans* [1]. Despite a lot of research, the upstream sensing part and more particularly the role of Ras1 and Ras2 in the different virulence traits remain elusive. In this paper, we will focus on the role of Cdc25, Ras1, and Ras2 for their contribution to several in vitro and ex vivo virulence characteristics.

Cdc25 consists of 1333 amino acids and contains a highly conserved C-terminal region that it is part of the GDP-GTP exchange factor domain towards Ras1 [12,13]. A *cdc25* mutant is still viable, although it has a slower growth rate compared to the WT strain and is unable to form hyphae on a serum-containing medium [14]. 

Ras1 consists of 291 amino acids and is important for both cell viability and virulence [15]. It is localized near the plasma membrane via its C-terminal motif and shows high similarity with its ortholog in *S. cerevisiae* [16,17,18]. Ras1 is very important for cAMP production by Cyr1 as a *ras1* mutant produces 20 times lower cAMP levels compared to a WT strain [18]. However, Ras1 is not involved in maintaining basal levels of cAMP in the cells, since *ras1* mutants still contain basal cAMP levels [15,18]. Due to its effect on cAMP levels and hence PKA activation, Ras1 is involved in the regulation of different virulence factors. A *ras1* mutant is unable to form hyphae on serum containing media and upon phagocytosis by macrophages [16,19]. As the *ras1* mutant is also sensitive to oxidative stress, this strain shows a reduced survival rate after coculturing with macrophages. The mutant shows a reduced virulence during a systemic infection in mice, with no hyphae but only yeast-like cells found in the kidneys [19,20]. However, a *ras1* mutant shows higher survival in the presence of neutrophils compared to the WT strain [21]. Furthermore, it has a reduced toxicity towards oral epithelial cells and a delayed apoptosis after addition of acetic acid and hydrogen peroxide [22,23,24]. 

Ras2 is an atypical Ras protein that shows low similarity in sequence with other Ras proteins [18]. However, it contains the typical CCIIT sequence for membrane anchoring, suggesting localization of this protein at the plasma membrane [18,25]. Ras2 has a negative effect on Cyr1 activity as a deletion of *RAS2* in a *ras1* background restores the cAMP levels to 30% of the WT level after glucose addition [18]. Furthermore, Ras2 seems to have a small effect on morphogenesis, but this is masked by Ras1 as deletion of *RAS2* in a *ras1* background worsens the hyphal defect [18]. The synergistic effect of Ras1 and Ras2 on morphogenesis and their antagonistic effect on cAMP levels suggest that these proteins affect the cell morphology both via a cAMP-dependent and independent mechanism [15]. A *ras2* mutant also shows lower resistance towards hydrogen peroxide and heavy metal Co^2+^ which is opposite to a *ras1* mutant [18].

As mentioned above, a lot is known about the role of Ras1 during virulence; however, we decided to include the *ras1* mutant in this study, since some of the previous experiments reported were performed in different background strains and previous studies indicated the importance of the background strain in regulatory networks [26]. Furthermore, we wanted to compare our results for Cdc25 and Ras2 with the results for Ras1. We suggest that, apart from Ras1, Cdc25 also has an effect on in vitro and ex vivo virulence characteristics via both a PKA-dependent and PKA-independent mechanism. We show that a deletion of *CDC25* or *RAS1* results in less toxicity towards oral epithelial cells. This is expected as these deletions result in a morphogenesis defect and hyphal formation is important during the invasion of epithelial cells. However, the toxicity towards cervical cells increases in a *cdc25* and *ras2* mutant while decreases in a *ras1* mutant compared to the WT. Furthermore, we found that a *cdc25* and *ras1* mutant have a higher survival rate in the presence of primary macrophages compared to the WT. This indicates a negative effect of PKA on this virulence trait. 

## 2. Materials and Methods

### 2.1. Culturing Conditions of C. albicans

Cells were grown at 30 °C in YP or SC medium supplemented with 100 mM glucose, unless stated otherwise. YP medium contains 1% yeast extract and 2% bacteriological peptone. SC medium contains 0.079% complete CSM (MP biomedicals, Santa Ana, CA, USA), 0.17% yeast nitrogen base without amino acids or ammonium sulfate ((NH_4_)_2_SO_4_; Difco), and 0.5% (NH_4_)_2_SO_4_. 1.5% agar was added to obtain solid medium. Both YP and SC medium could be supplemented with 5 mM, 50 mM, or 100 mM glucose, fructose, galactose, or glycerol. During the transformations, nourseothricin (NAT) was added to the medium to a final concentration of 200 µg/mL to select for cells which contained the NAT marker. To flip out this marker, cells were grown at 30 °C on YP 100 mM maltose overnight. Afterwards, the culture was plated for single colony and cells were restreaked on NAT containing solid medium to check for growth. RPMI 1640 with L-glutamine (Sigma-Aldrich, Saint Louis, MO, USA) was used to test the growth of the strain in physiological relevant medium. This medium was buffered with 0.165 M morpholinepropanesulfonic acid (MOPS) to pH 7. For the filamentation check, different hyphal inducing solid media were used: YP with 100 mM glucose, YP with 100 mM glucose and 10% fetal bovine serum (FBS), Spider (1% nutrient broth, 1% mannitol and 0.2% potassium phosphate (KH_2_PO_4_)), SLAD (0.17% yeast nitrogen based without amino acids and (NH_4_)_2_SO_4_, 6.6 × 10^−4^% (NH_4_)_2_SO_4_ and 100 mM glucose), SLD (0.17% yeast nitrogen based without amino acids and (NH_4_)_2_SO_4_, 0.5% (NH_4_)_2_SO_4_ and 0.1%. glucose), and Lee (0.5% (NH_4_)_2_SO_4_, 0.02% magnesium sulfate (MgSO_4_), 0.025% KH_2_PO_4_, 0.5% sodium chloride (NaCl), 7.16 × 10^−2^% ornithine, 0.05% alanine, 0.3% leucine, 0.1% lysine, 0.05% phenylalanine, 0.05% proline, 0.05% threonine, 0.01% methionine 0.1% biotin, 62.5 mM glucose).

### 2.2. Culturing Conditions of L-929 Cells

The supernatants of L-929 cells were collected and filter sterilized after culturing the cells for 1 week to obtain L-conditioned medium. Cells were incubated at 37 °C and 5% CO_2_ in L-cell medium (Iscove’s Modified Dulbecco’s medium (IMDM; Gibco, Waltham, MA, USA), 10% heat-inactivated Fetal calf serum (FCS; Gibco), 50,000 U Penicillin-Streptomycin (Gibco), 1% non-essential amino acids (Gibco), and 1% Sodium pyruvate (Gibco). 

### 2.3. Cultivation of Primary Bone Marrow Derived Macrophages

Bone marrow from female C57BL/6 mice of 12 weeks was differentiated into macrophages by using L-conditioned medium. Femur and tibia were removed from mice and bone marrow cells were collected. Next, the bone marrow-derived macrophages (BMDMs) were cultured for 10 days in differentiation medium (IMDM medium, 10% heat-inactivated FCS (Gibco), 50,000 U Penicillin-Streptomycin (Gibco), and 30% L-conditioned medium) before they were used during experiments. 

### 2.4. Cultivation of HeLa Cells

A vial of HeLa cells was taken from the liquid nitrogen tank and the cells were thawed for 2 min at 37 °C. The vial was cleaned with 70% ethanol and the content was added to 9 mL growth medium (Dulbecco’s Modified Eagle medium (DMEM; Gibco) supplemented with 1× Glutamax (Gibco), antibiotics (Gibco) and 10% FCS). The cells were centrifuged for 10 min at 100 g and the supernatants was discarded. Then, 2 mL of pre-warmed fresh growth medium was added to the cells and the cells were transferred to a culture flask prefilled with 20 mL growth medium. After three days, the cells were further cultured by removing the medium and washing the cells two times with 1× PBS (Gibco). Next, 1× trypsin was added to the cells and incubated for 5 min at 37 °C. Afterwards, 15 mL of fresh growth medium was added to the cells and the cells were counted by making use of a counting chamber. Cells were seeded at 100,000 to 200,000 cells/mL and incubated at 37 °C and 5% CO_2_.

### 2.5. Cultivation of TR146 Cells

A vial of TR146 cells was taken from the liquid nitrogen tank and the cells were thawed for 2 min at 37 °C. The vial was cleaned with 70% ethanol and the content was added to 9 mL of TR146 medium (DMEM/Nutrient Mixture F-12 (Gibco) supplemented 10% heat-inactivated FCS and 50,000 U Penicillin-Streptomycin (Gibco)). The cells were centrifuged for 10 min at 100 g and the supernatants was discarded. Then, 2 mL of fresh TR146 medium was added to the cells and the cells were transferred to a culture flask prefilled with 20 mL TR146 medium. After three days, the cells were further cultured by removing the medium and washing the cells two times with 1× PBS (Gibco). Next, 5× trypsin was added to the cells and this was incubated for 5 min at 37 °C. Afterwards, 15 mL of fresh TR146 medium was added to the cells and the cells were counted by making use of a counting chamber. Cells were seeded at 750,000 to 2,000,000 cells/flask and incubated at 37 °C and 5% CO_2_.

### 2.6. Construction of the Deletion Collection

A CRISPR genome editing method with a cloning free approach to obtain different components of this system was used [27]. Fragment A was amplified via PCR from plasmid pADH110 and contained the second part of the NAT marker. Fragment B was amplified from plasmid pADH147 and contained a specific gRNA for the gene of interest. To obtain fragment C, fragment A and B were linked with each other via stitching PCR. The CAS 9 gene and the first part of the NAT marker were present on plasmid pADH99 which was cut at the MssI site to obtain a linear fragment. Both fragment C and the cut pADH99 plasmid were transformed into the strain of interest to delete a gene. The obtained colonies were checked via PCR and the ones with the deleted gene were cultured in YP maltose (100 mM) to lose the *CAS9* gene and the NAT marker. The transformation was done by using the Gietz method. At least three independent transformants of each strain were created to work further with (Table 1).

### 2.7. Growth Curve

Strains were grown overnight in YP glucose (100 mM). Cells were collected and washed twice with sterile Milli-Q water. Samples were diluted to OD_600_ 0.1 in sterile Milli-Q water. Then, 20 µL of the cell suspension was added to 180 µL of SC medium supplemented with the indicated carbon source or RMPI medium in a sterile 96-well plate. The OD_600_ was measured every half hour for 72 h in a Multiskan (Thermo Fisher, Waltham, MA, USA). Data analysis was performed in Graphpad Prism by using an ANOVA test with Bonferroni correction.

### 2.8. Spot Assay

Strains were grown overnight in YP glycerol (100 mM). Cells were collected and washed twice with sterile Milli-Q water. Samples were diluted to OD_600_ 0.1 in sterile Milli-Q water. A three-time dilution series of 1/10 was made to have in total four dilutions of one culture. Then, 5 µL of each dilution was spotted on SC medium supplemented with glucose, fructose, galactose, and glycerol. The plates were incubated at 30 °C and pictures were taken after 24 h and 48 h.

### 2.9. Macrophage Survival Experiment

BMDMs were seeded in a Nunc™ F96 MicroWell™ plate (1 × 10^5^ cells/well) in differentiation medium and infected with *C. albicans* cells (1 × 10^4^ cells/well) in triplicates. Three hours after co-culturing, 4% Triton-X 100/PBS solution was added to lyse the macrophages. *C. albicans* cells were collected, plated, and colony forming units (CFUs) were counted. Data analysis and statistical analysis were performed in Graphpad Prism by using an ANOVA test with Bonferroni correction.

### 2.10. HeLa Cell Toxicity Assay

HeLa cells were collected and counted as described earlier. The cells were diluted to a concentration of 10^5^ cells/mL in growth medium, 100 µL of this suspension was added to the wells of a Nunc™ F96 MicroWell™ plate (Thermo fisher), and the plate was put for 24 h at 37 °C and 5% CO_2_. The next day, *C. albicans* cells of an overnight culture were collected and were washed three times with 1× PBS. The OD_600_ was measured, and the cells were diluted to a final concentration of OD_600_ 10. For the experiments with the additional dibutyryl cAMP (Sigma), cells were diluted in growth medium supplemented with the specific dibutyryl cAMP concentrations. A measure of 10 µL of the cell suspension was added to the HeLa cells and incubated for 24 h at 37 °C and 5% CO_2_. At day three of the experiment, the cytotoxicity of the *C. albicans* cells towards the HeLa cells was measured by making use of the CyQUANTUM LDH Cytotoxicity Assay Kit (Invitrogen, Waltham, MA, USA). First, 10 µL of lysis buffer was added to the positive control wells which only contained HeLa cells which was incubated for 45 min 37 °C and 5% CO_2_. Next, 50 µL supernatants of the wells was transferred to a new 96-well plate, 50 µL of reaction buffer was added, and the mixtures were mixed by tapping gently to the wells. After 30 min of incubation in the dark at room temperature, 50 µL of stop solution was added to the wells and the absorbance was measured at 490 nm and 680 nm by making use of the Synergy (BioTek, Winooski, VT, USA). Data analysis and statistical analysis was performed in Graphpad Prism by using an ANOVA test with Bonferroni correction.

### 2.11. TR146 Cell Toxicity Assay

TR146 cells were collected and counted as described earlier. The cells were diluted to a concentration of 105 cells/mL in TR146 medium, 100 µL of this suspension was added to the wells of a Nunc™ F96 MicroWell™ plate (Thermo fisher), and the plate was put for 24 h at 37 °C and 5% CO_2_. The next day, *C. albicans* cells of an overnight culture were collected and were washed three times with 1× PBS. The OD_600_ was measured, and the cells were diluted to a final concentration of OD_600_ 1. Then, 10 µL of the cell suspension was added to the TR146 cells and incubated for 24 h at 37 °C and 5% CO_2_. At day three of the experiment, the cytotoxicity of the *C. albicans* cells towards the TR146 cells was measured by making use of the CyQUANTUM LDH Cytotoxicity Assay Kit (Invitrogen). First, 10 µL of lysis buffer was added to the positive control wells, only containing TR146 cells, which were incubated for 45 min 37 °C and 5% CO_2_. Next, 50 µL supernatants of the wells was transferred to a new 96-well plate, 50 µL of reaction buffer was added, and the mixtures were mixed by tapping gently to the wells. After 30 min of incubation in the dark at room temperature, 50 µL of stop solution was added to the wells and the absorbance was measured at 490 nm and 680 nm by making use of the Synergy. Data analysis and statistical analysis was performed in Graphpad Prism by using an ANOVA test with Bonferroni correction.

## 3. Results

### 3.1. Making Deletion Strains

The different deletion strains used in this study were made via the CRISPR Cas9 system of the Hernday lab [28]. We were able to delete *CDC25*, *RAS1*, *RAS2*, *EFG1*, and *CPH1* and we obtained three biological transformants per strain. 

### 3.2. Deletion of CDC25 and RAS1 Affects Growth on Both Liquid and Solid Medium

To check the growth ability of the single mutants, both a growth curve in liquid and a spot assay on solid medium at 30 °C was performed. For the growth curve in liquid medium, the strains were tested on SC medium supplemented with 5 mM glucose, 100 mM glucose, 100 mM fructose, and 100 mM galactose, and RPMI medium. On high sugar concentrations, both a *cdc25* and *ras1* mutant showed a significant reduced growth rate compared to the WT strain and this growth defect was more severe for the *ras1* mutant than for the *cdc25* mutant (Figure 1A–C). However, on 5 mM glucose and RPMI medium all strains grew equally well as no significant difference was found (Figure 1D,F). No growth difference was observed when cells were grown on SC medium without carbon source (Figure 1E). The *ras2* mutant showed the same growth capacity as the WT strain on all tested conditions (Figure 1). 

Using a spot assay, we assessed the growth of the mutants on solid SC medium without or with a carbon source (5 mM glucose, 100 mM glucose, 100 mM fructose, and 100 mM galactose). The obtained results slightly differ with the ones from the liquid growth curve. Both the *cdc25* and *ras1* mutant showed a reduced growth compared to the WT strain on all tested media (Figure 2). A growth defect of the *ras1* mutant on SC 5 mM glucose was observed while there was no significant difference in liquid medium. The *ras2* mutant showed the same growth rate as the WT (Figure 2).

### 3.3. Toxicity of C. albicans Cells towards HeLa Cells Does Not Depend on Cdc25

From literature, it is known that a *cdc25* and *ras1* deletion strain are not able to filament upon addition of serum which we confirmed with our strains (Appendix A) [19]. Since filamentation is an important factor to inflict damage to host cells, we wanted to check if these mutants had a problem with toxicity towards HeLa cells. It was already shown that a *ras1* mutant has a decreased toxicity towards oral epithelial cells. However, since that mutant was obtained in a CAI4 background and we used cervical cells, we also included the *ras1* mutant in our study [24,29]. We showed that a *ras1* mutant indeed has a reduced toxicity towards these cells, but a *cdc25* mutant showed increased toxicity while it has a defect in filamentation (Figure 3A, Appendix A). This observation was unexpected since it was thought that the effect of Ras1 on cell damaging would work via the PKA pathway. However, if this was the case, a same phenotype should be observed for a *cdc25* and *ras1* mutant and this was not the case. Furthermore, a *ras2* mutant also caused a higher toxicity towards the HeLa cells (Figure 3A). 

To check if this decreased cell damage by a *ras1* mutant was caused by a lack of cAMP production, we repeated this experiment and added different concentrations of dibutyryl cAMP, but none of the tested concentrations could rescue the decreased cell toxicity of the *ras1* mutant (Figure 3B). This suggests that the effect of Ras1 on the cell damage of HeLa cells works independently of the cAMP/PKA pathway.

As we obtained these peculiar results, we wanted to further investigate which mechanism is important during infection of HeLa cells. Hence, we investigated the role of two downstream pathways of Ras, namely the PKA pathway and the MAPK pathway. The MAPK pathway is important for sensing and responding to specific environmental conditions, like serum, cell wall damage or osmotic stress. Different routes of the MAPK pathway are described for *C. albicans*, such as the Cek1/Cph1 route [30]. We investigated the role of the PKA and MAPK pathway by making use of an *efg1* mutant and a *cph1* mutant. *EFG1* and *CPH1* encode transcription factors activated by PKA and MAPK, respectively. This would allow us to see the role of these two transcription factors downstream of their specific pathway during virulence towards HeLa cells. We observed that an *efg1* mutant showed a decreased toxicity while a *cph1* mutant had an increased toxicity towards HeLa cells (Figure 4A). In a next step, we also investigated the effect on HeLa toxicity of different double deletion strains: *cdc25 efg1*, *cdc25 cph1*, *ras1 efg1*, and *ras2 cph1*. We showed that both the *cdc25 efg1* and *ras1 efg1* mutant had a significant reduced toxicity compared to the WT, while both *cdc25 cph1* and *ras2 cph1* showed no difference in toxicity (Figure 4B).

### 3.4. Cdc25 and Ras1 Are Involved in Toxicity towards Oral Epithelial Cells

To check if different mechanisms played a role in different host niches, we repeated the cytotoxicity experiment with TR146 cells, which is an oral epithelial cell line. Indeed, different niches gave different results. We showed that a *cdc25* mutant caused less toxicity towards TR146 cells, while it had a higher toxicity towards HeLa cells (Figure 4A and Figure 5A). Furthermore, a *ras2* and *cph1* mutant showed the same toxicity as the WT towards TR146 cells, while they had an increased toxicity towards HeLa cells (Figure 4A and Figure 5A). Finally, a *ras1* and *efg1* mutant showed the same toxicity towards both tested cell types indicating that Ras1 and Efg1 are essential components during host toxicity independently from the niche and conditions in these niches (Figure 4A and Figure 5A). We also investigated the effect of the double mutants on toxicity towards oral epithelial cells. Both a *cdc25 efg1* and a *ras1 efg1* mutant showed a drastic decreased toxicity compared to the WT (Figure 5B). The *cdc25 cph1* mutant also showed a decreased toxicity compared to the WT, however, this was less pronounced compared to the other mutants (Figure 5B). 

### 3.5. Deletion of CDC25 or RAS1 Results in a Higher Survival of C. albicans in the Presence of Macrophages

Filamentation is also an important virulence trait to escape macrophages, which constitutes an important part of the immune defense against *C. albicans* [31]. Since *cdc25* and *ras1* mutants are hypofilamentous, we suggested that they would have difficulties to survive and thrive during a coculture with macrophages since hyphae are necessary to counteract these immune cells [19,31]. As it is suggested that the effect on morphogenesis works both in a PKA-dependent and PKA-independent manner, we investigated the effect of Cdc25, Ras1, and Ras2 on the survival of *C. albicans* in the presence of macrophages. A macrophage coculturing experiment with a *ras1* mutant was reported previously. However, this study used a CAI4 *C. albicans* background, which can give some background effects in virulence assays [19,29]. Since our mutants were made in the SC5314 strain and we wanted to test the effect of Cdc25 and Ras2 on survival in the presence of macrophages, we also included the *ras1* mutant in this experiment. Interestingly, we obtained different results with our experiment compared to the reported study by the group of Schröppel [19]. We showed that a *cdc25* and *ras1* mutant had an increased survival in the presence of macrophages compared to the WT (Figure 6). This indicates that other factors than morphogenesis are involved in macrophage escape and are affected in these mutant strains. A *ras2* mutant showed no difference with the WT (Figure 6). 

## 4. Discussion

The Ras1/cAMP/PKA pathway is important for cell growth and virulence in *C. albicans* [1]. Such an important and central pathway requires extensive research to check the effect of its different components, such as for example Ras1, on virulence. However, much of this pathway remains unknown, such as the role of its upstream activator, Cdc25, and the other Ras component in the cell, Ras2. Therefore, we wanted to see the effect of Cdc25 and Ras2 on growth and virulence of *C. albicans* towards mammalian cells and thus see which virulence factors in these settings are regulated via PKA activation and which are regulated via other mechanisms. We also included our *ras1* mutant in this study to have a complete picture so that we can compare the different strains in one experiment. Furthermore, most of the experiments with *ras1* mutants found in literature are performed with a CAI4 background strain which by itself already has different phenotypes compared to the SC5314 background strain [29]. 

Both Cdc25 and Ras1 are important for a normal growth rate on all tested carbon sources. Deletion of *CDC25* results in a significant decreased growth compared to the WT (except for 100 mM fructose), and a *ras1* mutant shows an even more significant affected growth. However, a growth defect of the *cdc25* and *ras1* mutant on 5 mM glucose and RPMI is absent compared to higher sugar concentrations. This can be explained by the fact that cells generally grow slower at these low sugar concentrations. Therefore, the growth defect is not clearly pronounced, and it is harder to distinguish the growth of the mutant compared to the growth of the WT strain. Since the growth defect is observed in both a *cdc25* and *ras1* mutant, this defect can be caused by inactivation of the PKA pathway. The difference between the growth rate of these two mutants can be explained by the fact that Cdc25 activates Ras1, but basal activated Ras1 levels are also present without activation by Cdc25. We also see a significant growth defect of the *cdc25* and *ras1* mutant on carbon sources such as fructose, galactose, and glycerol during growth on solid medium [1,2,4]. These results are unexpected as these three carbon sources do not activate PKA. Therefore, we did not expect an effect of a *CDC25* or *RAS1* deletion on the growth on fructose, galactose, and glycerol as we assign the growth defect of these two mutant strain to inactivation of the PKA pathway. However, we also show that the *cdc25* and *ras1* mutant have a different growth pattern compared to the WT on solid SC medium without a carbon source. This indicates that the growth defect of the *cdc25* and *ras1* mutant is not caused by inactivation of PKA due to these deletions otherwise we would only observe a growth defect on carbon sources that are able to activate PKA. It is possible that these two proteins influence cell growth independently of PKA and thus we see this growth defect during all tested conditions. 

Different steps are involved in damaging human epithelial cells by *C. albicans*: adhesion, penetration, and lysis [32]. Since PKA regulates different genes involved in this process, it is suggested that inhibiting this pathway can result in a reduced toxicity. This would indicate that deletion of *CDC25* and *RAS1* also decreases the toxicity. However, this is not observed; the *cdc25* mutant shows an increased toxicity, while the *ras1* mutant shows decreased toxicity. As these two mutants show antagonistic phenotypes and the addition of extra dibutyryl cAMP does not rescue this *RAS1* deletion phenotype, it suggests that this effect is independent from PKA. In contrast, an *efg1* mutant, a downstream transcription factor of PKA, a *cdc25 efg1* mutant, and a *ras1 efg1* mutant show the same reduced toxicity phenotype as the *ras1* mutant [33]. To conclude, we hypothesize that the phenotype of the *ras1* mutant is not caused by inactivation of PKA. Since Ras1 works upstream and Efg1 downstream of PKA, we obtain similar phenotypes with these mutants, but the molecular mechanisms behind can be completely different [33]. 

Since we hypothesize that PKA is not the main pathway in the toxicity towards HeLa cells, we investigated the role of the MAPK pathway, as this pathway is also involved in virulence [6]. Our data suggest that Cph1 has a negative effect on epithelial damage as a *cph1* mutant shows an increased toxicity compared to the WT. Recently, it was shown that activation of Cph1 results in the unmasking of β-(1,3)-glucan which results in recognition by the HeLa cells [34,35]. It is possible that due to the deletion of *CPH1,* less β-(1,3)-glucan unmasking occurs and consequently *C. albicans* has an increased toxicity towards these epithelial cells as the pathogen is not recognized. Furthermore, to influence the pathogen, the epithelial cells need to produce different cytokines to recruit immune cells which will clear the infection [34]. In our experimental setup, only HeLa cells were present, so the difference in β-(1,3)-glucan masking between the WT and mutant could not result in a different amount of recruited immune cells as they were not present. Interestingly, the *cdc25*, *ras2*, and *cph1* mutants show the same phenotype in cell damage towards HeLa cells. It is possible that these three proteins are working in the same mechanism that negatively regulates toxicity. We hypothesize that Cdc25 activates Ras2 which on its turn is responsible for the activation of the MAPK pathway and finally Cph1, instead of Ras1. This can explain the similar phenotype of the *cdc25*, *ras2,* and *cph1* mutant towards cervical cell toxicity. If this is indeed the mechanism, we hypothesize that Cdc25 is activated via multiple stimuli which results in a specific conformational change dependent on the stimulus. A specific conformation can result in either activation of Ras1 or Ras2 which consequently leads to the activation of the PKA or MAPK pathway, respectively. The double mutant *cdc25 cph1* shows the same phenotype as the WT which is unexpected. On the other hand, the double mutant *ras2 cph1* shows the same phenotype as the *ras2* and *cph1* single mutants which strengthens our hypothesis that these two proteins work in the same mechanism. However, previous research showed that a *ras1* and *cph1* mutant give the same morphogenesis defect so they assume that Ras1 is the upstream activator of Cph1 [1]. Another explanation for the increased toxicity towards cervical cells in the *ras2* mutant can be found in the effect of Ras2 on cAMP levels. Literature shows that Ras2 has a negative effect on the cAMP levels in the cell [18]. By deleting *RAS2*, higher cAMP levels can be present which can result in overactivation of PKA and consequently a higher toxicity towards cervical cells.

Glucose is present in high concentrations in the mouth of the human body after food uptake [36]. Therefore, glucose can be important for *C. albicans* cells present in the oral cavity as it is the main and preferred energy source for this pathogen, and it can activate the PKA pathway. Furthermore, glucose activation of the PKA pathway in this niche is a key factor since this pathway regulates different genes involved in epithelial cell damage [32]. Both a *cdc25* and *ras1* mutant show a decreased toxicity in comparison to the WT. These findings are strengthened by the phenotype of the *efg1* mutant which is already shown in literature [37]. Due to the deletion of *EFG1*, the expression of *SAP1* and *SAP3*, genes encoding proteases involved in the degradation of specific host cell components, is reduced which results in a decreased toxicity towards oral epithelial cells [37]. Therefore, the reduced toxicity observed in the *cdc25* and *ras1* mutant strains is probably due to inactivation of PKA and consequently Efg1 which results in reduced expression of *SAP1* and *SAP3*. Probably, other genes encoding proteins involved in cell damage are affected by these deletions and contribute to these phenotypes. This hypothesis is strengthened by the phenotype of the *cdc25 efg1* and *ras1 efg1* double deletion strain which shows that Efg1 has a function downstream of Cdc25 and Ras1. The difference in toxicity between the *cdc25* and *ras1* mutants is quite big, indicating that basal Ras1 activation seems sufficient for partial toxicity. However, it is also possible that Ras1 has an extra function in the toxicity towards TR146 cells besides Cyr1 activation, such as a cAMP-independent effect on morphogenesis of Ras1 as described earlier in literature [18]. Therefore, due to inhibition of PKA and these possible extra functions, the *ras1* mutant has a more drastic decreased toxicity compared to the *cdc25* mutant. In addition, the MAPK pathway does not seem to play a major role in the infection of oral epithelial cells as the *cph1* mutant and the WT strain or the *cdc25* mutant and the *cdc25 cph1* mutant show the same toxicity. The same is true for the *ras2* mutant as Ras2 has a negative effect on cAMP levels, so its deletion does not result in PKA inactivation [18].

Different niches in the human body have widely differing chemical characteristics and nutrient composition, such as pH, the presence of glucose, or oxygen levels. This influences the activation of specific proteins and pathways, like the Rim101 or PKA pathway [1]. Efg1 is a stimulator of morphogenesis but under certain conditions with limited oxygen access and embedded in agar, Efg1 is a repressor of morphogenesis [11]. When we compare our toxicity results obtained using HeLa and TR146 cells, the significantly different results may originate from the fact that different media are used to culture the different cell lines. As these media contain different glucose concentrations, it is possible that the toxicity difference towards the different cell lines is caused by a difference in glucose concentration. Furthermore, it is possible that *C. albicans* receives specific stimuli from these oral epithelial cells, which are absent in cervical cells, and activate PKA. 

Macrophages are an important part of the innate immune system against *C. albicans*, so this fungus has different mechanisms to cope with these macrophages to survive. For example, they neutralize the ROS molecules inside immune cells and form hyphae to break out [31]. Since Ras1 is involved in morphogenesis, partially via activation of PKA, it is logical that a deletion of *RAS1* results in a decreased survival in the presence of macrophages compared to the WT, as is shown by the group of Schröppel [16,19]. However, this is not what we observed in our experiments. We showed that both the *cdc25* and *ras1* mutant have an increased survival during coculturing with macrophages compared to the WT strain. As both mutants give a similar phenotype, it is hypothesized that this effect is caused by inactivation of the PKA pathway. The main question is why we obtain different results compared to the other research group. First, it is possible that a difference in strain background between the two studies affects the results. We made the mutants in a SC5314 background strain whilst the other group made their strains in the CAI4 background strain. It is known from previous studies that this CAI4 background gives different phenotypes compared to the prototrophic SC5314 strain due to differences in *URA3* expression [29]. This indicates that the decreased survival of the *ras1* mutant in the CAI4 strain can be caused by the strain background and not by the deletion itself. A second explanation can be the use of different macrophages in the different studies. We used macrophages derived from bone marrow whilst the other group used inflammatory peritoneal exudate-derived macrophages [19]. Therefore, it is possible that these different types react differently on the presence of *C. albicans* cells.

The PKA pathway regulates different processes involved in macrophage infections, like morphogenesis and adhesion [15]. Therefore, as previously mentioned, the increased survival of the *cdc25* and *ras1* mutant is unexpected. However, several hypotheses and explanations, involving recognition, uptake, and survival within the immune cells, can be formulated. First, the pathogen needs to be recognized through PAMP-PRR interactions with the cell wall displaying the different PAMPs to be recognized. It is known that the cell wall components are partially regulated via the PKA pathway [38,39]. Therefore, it is possible that by deleting *CDC25* and *RAS1* and hence inactivation of PKA, the cell wall and its constituents are altered, resulting in lower fungal cell recognition. Secondly, these two mutants are not able to form hyphae which is important to escape from inside the macrophages [16,31]. However, this does not result in a decreased survival rate inside macrophages. The observed increase in survival rate of the two mutants can be explained by the fact that due to the inability to form hyphae, the cells are recognized to a lesser extent since cells with a difference in morphology also display different cell wall proteins, such as adhesins. Another explanation can be found in counteracting ROS molecules to survive inside these macrophages [40]. It is possible that PKA plays a negative role in this process. This would explain the increased survival we obtain in the *cdc25* and *ras1* mutant as they have no active PKA. It is likely that this is the main reason why we observe this increased survival rate of these two mutant strains compared to the WT. Previously, it was shown that deletion of *RAS1* results in resistance to oxidative and nonoxidative killing by neutrophils [21]. Therefore, probably, the *cdc25* and *ras1* mutant have a higher resistance towards the ROS molecules produced by the macrophages compared to the WT.

## Figures and Tables

**Figure 1 jof-09-00201-f001:**
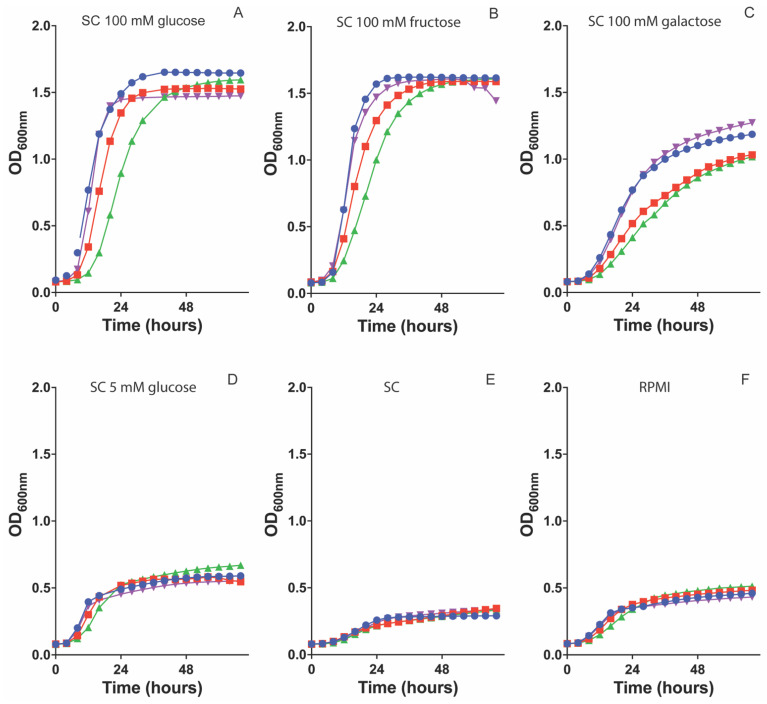
Deletion of *CDC25* and *RAS1* causes a reduced growth rate in liquid medium. The different strains were grown in different media: SC supplemented with 100 mM glucose (**A**), 100 mM fructose (**B**), 100 mM galactose (**C**), 5 mM glucose (**D**), SC (**E**), and RPMI (**F**). The OD_600_ was followed over time for 72 h. WT (●), *cdc25* (■), *ras1* (▲), and *ras2* (▼). The data represent the average of two independent experiments each consisting of three biological repeats and three technical repeats. Statistical analysis was done by using an ANOVA test with Bonferroni correction (Appendix A).

**Figure 2 jof-09-00201-f002:**
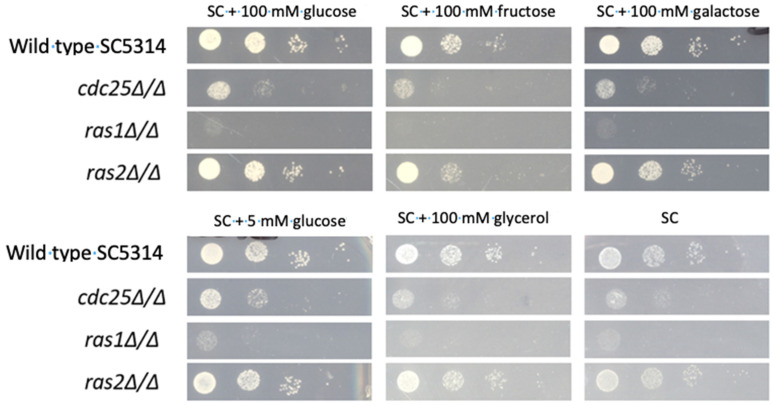
The *cdc25* and *ras1* mutant show a growth defect on solid medium. The different strains were grown on solid SC medium supplemented with 100 mM glucose, 100 mM fructose, 100 mM galactose, 5 mM glucose, or 100 mM glycerol. The plates were incubated for 24 h at 30 °C before pictures were taken.

**Figure 3 jof-09-00201-f003:**
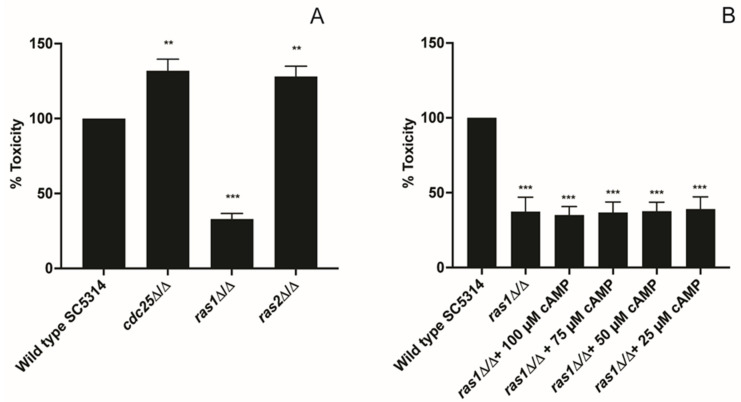
The cytotoxicity towards HeLa cells is influenced by Cdc25 and Ras1 but works independent from cAMP. HeLa cells were seeded at a concentration of 10,000 cells/well. *C. albicans* cells were added at OD_600_ 1. After an incubation of 24 h, the released amount of LDH from the HeLa cells was measured. (**A**) The toxicity of the WT was compared with a *cdc25*, *ras1*, and *ras2* mutant. (**B**) The effect of cAMP on the toxicity of the *ras1* mutant towards HeLa cells was tested. The values of the deletion strains are normalized relative to the WT strain which has 100% toxicity towards HeLa cells. The results show the average of three independent experiments each consisting of one biological repeat and three technical repeats with the error bars representing the SEM. Statistical analysis was done by using an ANOVA test with Bonferroni correction. A significant difference was found when ** *p* = 0.0021 and *** *p* = 0.0002.

**Figure 4 jof-09-00201-f004:**
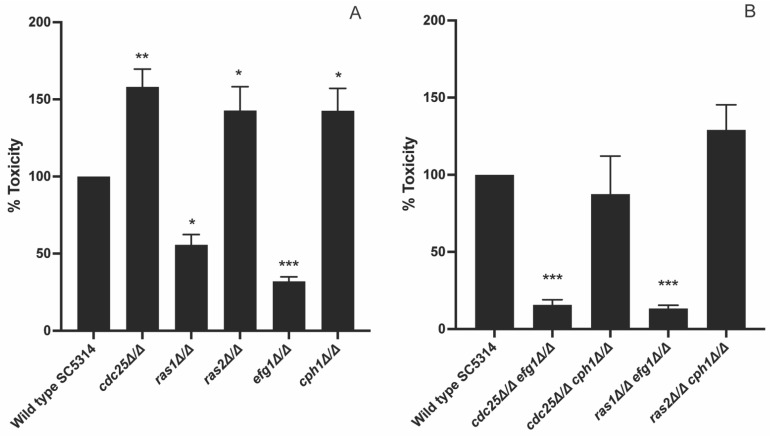
Efg1 has a positive effect while Cph1 has a negative effect on toxicity. HeLa cells were seeded at a concentration of 10,000 cells/well. *C. albicans* cells were added at OD_600_ 1. After an incubation of 24 h the released amount of LDH from the HeLa cells was measured. (**A**) The toxicity of the WT was compared with a *cdc25*, *ras1*, *ras2*, *efg1*, and *cph1* mutant. (**B**) The toxicity of the WT was compared with a *cdc25 efg1*, *cdc25 cph1*, *ras1 efg1*, and *ras2 cph1* double mutant. The values of the deletion strains are normalized relative to the WT strain which has 100% toxicity towards HeLa cells. The results show the average of three independent experiments each consisting of one biological repeat and three technical repeats with the error bars representing the SEM. Statistical analysis as done by using an ANOVA test with Bonferroni correction. A significant difference was found when * *p* = 0.0332, ** *p* = 0.0021, and *** *p* = 0.0002.

**Figure 5 jof-09-00201-f005:**
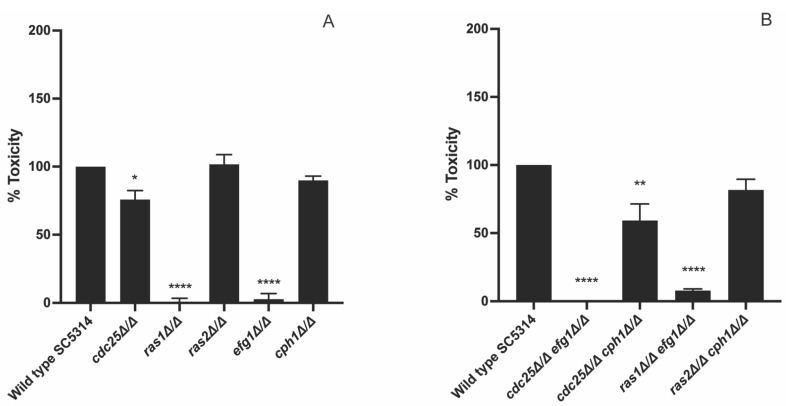
The Ras1/PKA pathway seems to be involved in toxicity towards oral epithelial cells. TR146 cells were seeded at a concentration of 10,000 cells/well. *C. albicans* cells were added at OD_600_ 0.1. After an incubation of 24 h, the released amount of LDH from the TR146 cells was measured. (**A**) The toxicity of the WT was compared with a *cdc25*, *ras1*, *ras2*, *efg1*, and *cph1* mutant. (**B**) The toxicity of the WT was compared with a *cdc25 efg1*, *cdc25 cph1*, *ras1 efg1*, and *ras2 cph1* double mutant. The values of the deletion strains are normalized relative to the WT strain which has 100% toxicity towards TR146 cells. The results show the average of three independent experiments each consisting of one biological repeat and three technical repeats with the error bars representing the SEM. Statistical analysis was done by using an ANOVA test with Bonferroni correction. A significant difference was found when * *p* = 0.0332, ** *p* = 0.0021, and **** *p* < 0.0001.

**Figure 6 jof-09-00201-f006:**
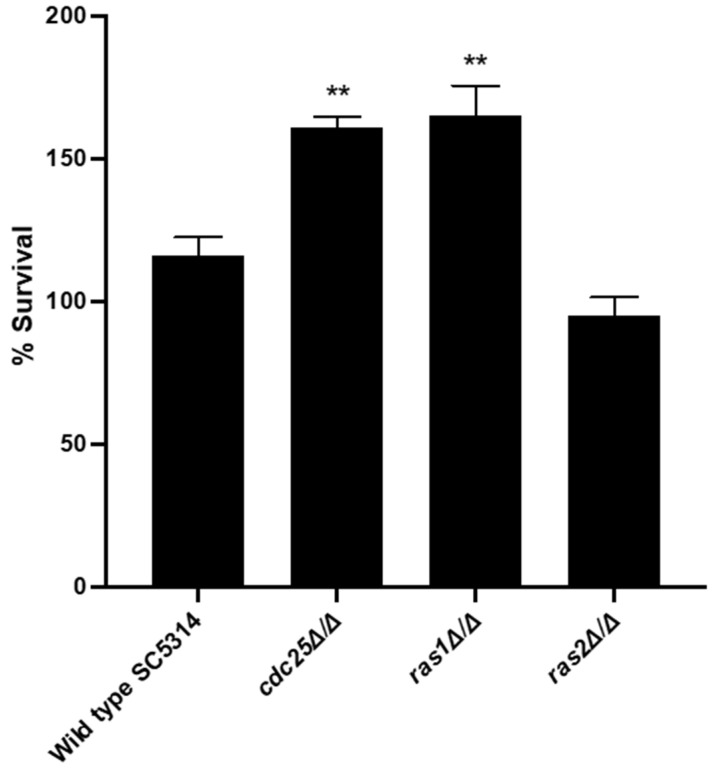
Cdc25 and Ras1 have a negative effect on the survival of *C. albicans* cells in the presence of macrophages. *C. albicans* cells were cocultured for three hours with BMDMs at 37 °C. Afterwards, the fungal cells were plated to check for their survival. The results show the average of three independent experiments each consisting of one biological repeat and three technical repeats with the error bars representing the SEM. Statistical analysis was done by using an ANOVA test with Bonferroni correction. A significant difference was found when ** *p* = 0.0021.

**Table 1 jof-09-00201-t001:** *C. albicans* strains used in this study.

Strains	Parent Strain	Source
SC5314		[28]
*cdc25* *Δ/* *Δ*	SC5314	This work
*ras1* *Δ/* *Δ*	SC5314	This work
*ras2* *Δ/* *Δ*	SC5314	This work
*efg1* *Δ/* *Δ*	SC5314	This work
*cph1* *Δ/* *Δ*	SC5314	This work
*cdc25* *Δ/* *Δ* *efg1* *Δ/* *Δ*	*cdc25* *Δ/* *Δ*	This work
*cdc25* *Δ/* *Δ* *cph1* *Δ/* *Δ*	*cdc25* *Δ/* *Δ*	This work
*ras1* *Δ/* *Δ* *efg1* *Δ/* *Δ*	*ras1* *Δ/* *Δ*	This work
*ras2* *Δ/* *Δ* *cph1* *Δ/* *Δ*	*ras2* *Δ/* *Δ*	This work

## Data Availability

Not applicable.

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
