# Peer review of "The Cdc25 and Ras1 Proteins of Candida albicans Influence Epithelial Toxicity in a Niche-Specific Way"

_jof, 2023, doi:10.3390/jof9020201_

Round 1

Author Response

  1. The authors should show the wild-type and mutant growth curves in SC liquid medium as a reference for the Fig.1 as they did for the solid medium in Figure 2.

We added a graph presenting strains growing on SC medium without carbon source.

  1. In Fig S1, ras2∆/∆ shows a filamentation defect on SLD medium. Is this a true phenotype for the ras2∆/∆ mutant?

This is a true phenotype and we added this observation in the figure legend.

  1. Looking at Fig. 2 (SC plate), it appears that the, ras1∆/∆ mutant grows slowly or does not grow. It might be possible that this mutant may not grow under the conditions employed to grow either HeLa or TR146 cell lines. I'm wondering if this severe growth defect is why it has significantly less toxicity towards host cells in Figs. 3 and 5. It is difficult to conclude that fungal cells are not toxic to the host in the absence of fungal growth. Is it possible for the authors to demonstrate that indeed the ras1∆/∆ mutant can grow under these host growth conditions?

This experiment is not feasible. The co-culturing of mammalian cells and fungal cells happens at 37°C and 5% CO2, but we cannot measure growth at 37°C due to possible hyphae formation. However, if we look at the growth curves at 30°C, we do not observe a major difference in growth between these strains. Therefore, we do not hypothesize that a difference in toxicity towards mammalian cells is caused by a difference in growth of the fungal cells.

  1. Lines 383-384 authors suggest that cAMP is required for yeast cell growth and that the slow growth phenotype of cdc25∆/∆ and ras1∆/∆ mutants is due to potentially low cAMP levels. This can be tested by running a growth experiment in the presence of cAMP, as the authors have nicely demonstrated in Fig. 3, to see if the addition of cAMP affects the cytotoxicity of the ras1∆/∆. If authors perform a growth experiment in the presence of exogeneous cAMP, they will be able to support their conclusion that "it is possible that these two proteins influence cell growth independently of PKA, and thus we see this growth defect during all tested conditions" lines 394-396.

We removed this hypothesis from the main text.

Reviewer 2 Report

Protein kinase A (PKA) pathway can be activated via the addition of glucose, which is related to the virulence factor in "Candida albicans". The activation involves at least two proteins (Cdc25 and Ras1), but it is not clear if these proteins also affect virulence independently of PKA. Then, we hypothesized and investigated here the role of Cdc25, Ras1, and Ras2 (the atypical Ras family in "C. albicans") for different in vitro and ex vivo virulence characteristics, using the deletion strategies of these genes. The study suggested that deletion of Cdc25 and Ras1, in contrast to Ras2, results in less toxicity towards oral epithelial cells, because these deletions result in a morphogenesis defect and hyphal formation is important during the invasion of epithelial cells. The authors reported evidence that, apart from Ras1, Cdc25 also has an effect on "in vitro" and "ex vivo" virulence characteristics via both a PKA-dependent and -independent mechanism. In my opinion, the authors have addressed most of my concerns appropriately, but I have a few minor points to make:

1. The authors need to revise all the abbreviations presented in the manuscript, carefully. some examples: PKA, Cdc25, Ras1, Cph1, and many others.

2. Statistical analysis details can be included in Materials and Methods. Maybe in the subsection.

3. 173–174, 205-206, and 222-223 lines Data analysis was performed in GraphPad Prism. The authors should present details of the analysis (ANOVA, for example).

4. In may legend of figures, thers is "A significant difference was found when p≤0.05*". However, statistical differences between some groups were identified with "***" or "****". Include details in the legends.

5. See 302 line. There is a problem in the sentence.

6. 509-520 lines. These details at the end of the discussion section confuse me, and they should be addressed when cited. 

Author Response

  1. The authors need to revise all the abbreviations presented in the manuscript, carefully. some examples: PKA, Cdc25, Ras1, Cph1, and many others.

We revised all abbreviations

  1. Statistical analysis details can be included in Materials and Methods. Maybe in the subsection.

The statistical analysis of each experiment is included in the materials and methods.

  1. 173–174, 205-206, and 222-223 lines Data analysis was performed in GraphPad Prism. The authors should present details of the analysis (ANOVA, for example).

We included the details of the statistical analysis in the text.

  1. In may legend of figures, thers is "A significant difference was found when p≤0.05*". However, statistical differences between some groups were identified with "***" or "****". Include details in the legends.

We included the details about the significant difference in the legends.

  1. See 302 line. There is a problem in the sentence.

We removed this sentence.

  1. 509-520 lines. These details at the end of the discussion section confuse me, and they should be addressed when cited.

We rephrased this part of the discussion to make it better understandable.

Reviewer 3 Report

This manuscript talked about the function of Cdc25 and Ras1 during the interaction with host cells. They made series KO cells in SC5314 background and used several host cell lines to test the toxicity caused by these Candida cells. This work may supply new ideas of the PKA pathway in virulence. However, this work was not well designed. Particularly, too many speculations were made with no sufficient surporting data.

1. The title mentioned that Cdc25 and Ras1 may influence host infection via both a PKA-dependent and PKA-independent mechanism in Candida Albicans. Among the whole manuscript, I didn’t find how this happens. How do these two proteins influence host infection through PKA? How do they infect host infection through other pathyways. They only tested one downstresm factor of PKA, Efg1, with no further investigation. It is quite weird that efg1 mutant show the same reduced toxicity phenotype as the ras1 mutant, but they conclude that these regulation is independent from PKA.

 2. In the whole paper, none necessary work uncovered the underlying mechanism. Several results focused on the differnt results in toxicity by deleting RAS1 and CDC25. How does this happen? Does they have any genetic interaction? Since ras1 and efg1, but not cph1, showed a similar result in toxicity, suggesting a potential linkage of these two genes, why not make a double mutaion of RAS1 and EFG1, or CPH1. In addition, the authors also showed these KO mutants have different survival when incubated with maropahges. They speculated that the cytokines may be different. Why not test some cytokines of macrophages when infected with these mutants.

3. The experiment  is not well desigened. We can only see a lot of graphs. Some images showing the incubation with host cells may be  necessary. The treating conditions used in this study is inadequate. They only used 1 infection ration and 1 treating time in this study. Since they found different results compared with previous studies, did they try other conditions.

4. The logic of this study is also poor. I only find they did some experiments using host cells, but actually I don' t know what they want to see, and which point they want to figure out. I suggest they can focus on one or two points and investigate it in detail. For example, How do these proteins influence infection indepent of PKA?

5. The organization of this manuscript need to be improved. The first result is not necessary. They already presented these information in the method section. The whole manuscript is based on these KO strains, defintely they got these strains. In addition, there is no figure 6 in the whole manuscript. The format of the gene name and strain name need to be consistent.

In general, this work is intesting, but the experiment must be re-desigened and the presentation need to be improved with reasonable logic.

Author Response

  1. The title mentioned that Cdc25 and Ras1 may influence host infection via both a PKA-dependent and PKA-independent mechanism in Candida Albicans. Among the whole manuscript, I didn’t find how this happens. How do these two proteins influence host infection through PKA? How do they infect host infection through other pathyways. They only tested one downstresm factor of PKA, Efg1, with no further investigation. It is quite weird that efg1 mutant show the same reduced toxicity phenotype as the ras1 mutant, but they conclude that these regulation is independent from PKA.

We think this title covers the main message of this paper. As the other referees have no problem with this title, we will not change it.

  1. In the whole paper, none necessary work uncovered the underlying mechanism. Several results focused on the differnt results in toxicity by deleting RAS1 and CDC25. How does this happen? Does they have any genetic interaction? Since ras1 and efg1, but not cph1, showed a similar result in toxicity, suggesting a potential linkage of these two genes, why not make a double mutaion of RAS1 and EFG1, or CPH1. In addition, the authors also showed these KO mutants have different survival when incubated with maropahges. They speculated that the cytokines may be different. Why not test some cytokines of macrophages when infected with these mutants.

We made additional double deletion strains and tested them during a cytotoxicity experiment to see their effect towards mammalian cells.

  1. The experiment is not well designed. We can only see a lot of graphs. Some images showing the incubation with host cells may be necessary. The treating conditions used in this study is inadequate. They only used 1 infection ration and 1 treating time in this study. Since they found different results compared with previous studies, did they try other conditions.

We optimized this experiment in the lab according as suggested in the manual of the LDH kit used in this study. Therefore, we think that it is not necessary to test multiple conditions.

  1. The logic of this study is also poor. I only find they did some experiments using host cells, but actually I don' t know what they want to see, and which point they want to figure out. I suggest they can focus on one or two points and investigate it in detail. For example, How do these proteins influence infection indepent of PKA?

We made additional double deletion strains and tested them during a cytotoxicity test to see which proteins work in the same pathway.

  1. The organization of this manuscript need to be improved. The first result is not necessary. They already presented these information in the method section. The whole manuscript is based on these KO strains, defintely they got these strains. In addition, there is no figure 6 in the whole manuscript. The format of the gene name and strain name need to be consistent.

We added figure 6 to the manuscript and revised all gene names and strains.

Round 2

Reviewer 3 Report

The manuscript can be accepted, but the result 1 should be moved to the method section.